# Effect of Methacrylic Acid Monomer on UV-Grafted Polyethersulfone Forward Osmosis Membrane

**DOI:** 10.3390/membranes13020232

**Published:** 2023-02-15

**Authors:** S. N. S. A. Aziz, M. N. Abu Seman, S. M. Saufi, A. W. Mohammad, M. Khayet

**Affiliations:** 1Faculty of Chemical and Process Engineering Technology, Universiti Malaysia Pahang, Lebuhraya Persiaran Tun Khalil Yaakob, Kuantan, Gambang 26300, Pahang, Malaysia; 2Earth Resources and Sustainability (ERAS) Centre, Universiti Malaysia Pahang, Lebuhraya Persiaran Tun Khalil Yaakob, Kuantan, Gambang 26300, Pahang, Malaysia; 3Chemical and Water Desalination Program, College of Engineering, University of Sharjah, Sharjah 27272, United Arab Emirates; 4Faculty of Engineering and Built Environment, Universiti Kebangsaan Malaysia (UKM), Bangi 43600, Selangor, Malaysia; 5Department of Structure of Matter, Thermal Physics and Electronics, Faculty of Physics, University Complutense of Madrid, Av. Complutense s/n, 28040 Madrid, Spain; 6Madrid Institute for Advanced Studies of Water (IMDEA Water Institute), Calle Punto Net No 4, Alcalá de Henares, 28805 Madrid, Spain

**Keywords:** UV grafting, PES membrane, forward osmosis, structural parameter

## Abstract

UV irradiation is one of the procedures that has been considered for membrane surface graft polymerization. It is commonly utilized for enhancing the wettability of polyethersulfone (PES) membranes. In this research study, the monomer methacrylic acid (MAA) was used for the UV grafting process of a commercial NF2 PES membrane for the preparation of a forward osmosis (FO) membrane. Three different monomer concentrations and three different UV irradiation times were considered. The intrinsic characteristics of both the surface-modified and pristine membranes were determined via a non-pressurized test method. Compared to the NF2 PES, the surface of the modified membranes was rendered more hydrophilic, as the measured water contact angle was reduced considerably from 65° to 32–58°. The membrane surface modification was also confirmed by the data collected from other techniques, such as atomic force microscopy (AFM), field emission-scanning electron microscope (FESEM) and Fourier-transform infrared spectroscopy–attenuated total reflectance (FTIR–ATR). Additionally, the modified membranes exhibited a greater water permeate flux (J_w_) compared to the NF2 PES membrane. In this study, the water permeability (A), solute permeability (B) and structural parameter (S) were determined via a two-stage FO non-pressurized test method, changing the membrane orientation. Compared to the FO pressurized test, smaller S values were obtained with significantly high A and B values for the two non-pressurized tests. The adopted method in the current study is more adequate for determining the intrinsic characteristics of FO membranes.

## 1. Introduction

During the last decade, an emerging interest has been shown in the osmotically driven membrane separation process, or forward osmosis (FO), because it does not require any applied hydrostatic pressure, so less specific energy consumption is required, and less fouling tendency is exhibited compared to pressure-driven membrane separation processes. However, FO membrane fouling still is considered to be one of the important issues affecting the performance of FO technology when treating wastewater. There are several works that have focused on the development of membranes with improved performance for FO applications [1,2,3,4,5,6,7,8]. These membranes were prepared by phase inversion [1,2,3], interfacial polymerization [2,4,5], the layer-by-layer method [6,7,8] and membrane grafting [9,10].

Polyether sulfone (PES) is one of the most commonly used popular polymeric materials in membrane engineering because of its good thermal and mechanical stability. However, PES membranes have a high fouling tendency caused by the adsorption of nonpolar solutes, hydrophobic particles or bacteria. Thus, PES membranes have been modified to increase the membrane’s antifouling property following a variety of methods such as membrane surface modification or bulk modification. Habitually, most researchers prefer surface modification due to its simplicity while keeping the original bulk properties of the membrane intact [11,12,13]. In this sense, UV grafting is the most frequently considered technique for the preparation of improved FO, nanofiltration (NF) and ultrafiltration (UF) membranes, as can be seen in Table 1. For instance, by applying the UV-grafting method, Seman et al. [14] compared the NF performance of two modified PES membranes using acrylic acid (AA) and N-vinyl-2-pyrrolidine (NVP) and found that the AA UV-grafted PES membrane displayed higher water permeability than the NVP UV-grafted PES membrane. The authors claimed that this result was probably due to the neutral charge of NVP, which facilitated the grafting procedure on the PES membrane surface, therefore forming a thicker layer and improving the hydrodynamic resistance but causing low permeability.

The first surface modification involving UV grafting of an UF PES membrane for FO application was recently proposed by Rahman and Seman [11]. A similar method was previously described for the surface modification of a NF PES membrane [18]. The NF PES membrane was modified via the UV-grafting method with an AA monomer observing an enhanced water permeate flux, four times greater than that of the unmodified membrane. The membranes were modified using different AA concentrations and a predetermined time; thus, different S values were obtained within the range of 5.9–12.5 mm. Up to now, there has been very few research studies on membranes that were UV-grafted for the FO process, especially those using commercial membranes as support. Many other monomers have yet to be examined for the development of improved surface-modified membranes for FO applications.

It must be pointed out that the evaluation of the membrane S parameter will differ depending on whether the applied method is pressurized or not [20]. The S value determined by a pressurized method such as reverse osmosis (RO) is always found to be higher than that by a non-pressurized one, since the water permeability (A) and solute permeability (B) obtained under the pressurized method are higher. This is because of the proportional relationship between A, B and S. The non-pressurized method normally involves two steps, as reviewed by Kim et al. [21] and first demonstrated experimentally by A. Aziz et al. [20].

The objective of the present research study was to investigate the FO performance of the UV-grafted NF2 PES membrane using a monomer MAA. The effects of the MAA concentration and irradiation time on the characteristics and FO performance of the modified membranes are investigated. Comparisons are carried out between the modified and unmodified NF2 PES membranes.

## 2. Materials and Methods

### 2.1. Materials

The NF PES membrane coded as NF2 by the manufacturer was purchased from Amfor Inc. (China). The MAA monomer and sodium chloride (NaCl) of ≥99.5% were obtained from Merck Chemicals, Darmstadt, Germany. Ultra-pure water was provided by a Milli-Q ultra-pure water system.

### 2.2. UV Irradiation

The UV light system (λ = 365 nm, Black-Ray B-100 Series) was purchased from UVP Ltd. The commercial NF2 PES membrane was modified by UV light irradiation in the presence of monomer solutions at different MAA concentrations (0.1 M, 0.5 M and 1.0 M) in deionized (DI) water. Before modification, the NF2 PES membrane was soaked overnight in a chiller and rinsed with deionized (DI) water to get rid of its protective layer. The membrane was placed in a petri dish filled with 50 mL of the MAA solution and left immersed for a given time (3, 5 and 10 min) under UV light. Subsequently, this membrane was rinsed with DI water to remove the unreacted compound and stored overnight in DI water for at least one day before characterization. The prepared membranes together with the modification conditions followed are tabulated in Table 2.

### 2.3. Morphological Characterization

The surface hydrophilicity of the membranes was evaluated by means of distilled water contact angle measurement using a goniometer telescope by Rame-Hart, Model 290. The degree of grafting (DG) of the modified membranes was determined from the measured weights of the membrane before and after UV grafting as follows:(1)$$\mathrm{DG}\left(\%\right)=\frac{W_{1}{\;-\;W}_{0}}{W_{1}}\;\times \;100\;$$
where W_1_ and W_0_ are the weights of the membrane after and before UV grafting, respectively. Both the surface and cross-section structure of the membranes were studied by field emission-scanning electron microscopy (JSM7800F Schottky FESEM, JEOL Ltd.). The surface topology of the membranes was examined with an atomic force microscope (AFM: scanning probe microscope, NX-10, Park Systems), as were the roughness parameters (the average surface roughness, Ra; the root mean square surface roughness, Rq; and the vertical distance between the highest peak and the lowest valley, Rz). Fourier-transform infrared spectroscopy (ATR-FTIR, Nicolet iS5 FTIR) was utilized to determine the functional groups of the membrane surface.

### 2.4. FO Membrane Performance

Figure 1 illustrates the laboratory scale FO system applied for the FO separation process. Basically, it consists of a CF042P-FO cell with an effective membrane area of 42 cm^2^ with 2.28 mm and 3.9 cm of depth and width, respectively. The system operates in a co-current mode with a double-head peristaltic pump (BT600-2J) as the feed and a permeate circulation flow rate of 200 mL/min. The volume of both the feed (DI) and permeate (draw solution: 1.0 M NaCl) was 1000 mL. The active layer (modified side of the membrane) was placed in the FO cell facing the feed solution (AL-FS). The weight change of the feed solution was recorded automatically by the data-logging system using the balance (AND FX-3000), and the concentration of the solutions was measured using an electrical conductivity meter (Eutech Instruments PC2700).

The water flux (J_w_) can be calculated by the following equation:(2)$$J_{w}=\frac{∆V}{A_{m}\;∆t}$$
where ΔV is the change in volume of the draw solution, Am is the effective membrane area, and t is the operating time.

The reverse solute flux from the draw solution to the feed can be obtained by the following equation [22]:(3)$${\;J}_{s}=\frac{∆C_{t}{\;V}_{t}}{A_{m}\;t}$$
where Ct and Vt are the concentration and volume of the feed solution, respectively.

Evaluations of the intrinsic parameters, water permeability (A), solute permeability (B) and structural parameter (S) were determined by means of a two-stage non-pressurized test. Initially, the NF2 PES membrane was wetted overnight in DI water. The experiment began by circulating ultra-pure water through both the feed solution (FS) and draw solution (DS) sides of the membrane in ALDS (active layer facing draw solution) orientation to ensure temperature equilibrium throughout the system. Once it was thermally stable, DS was replaced with 0.25M NaCl aqueous solution. The mass change was automatically recorded every 5 min using WinCT© software, and the electrical conductivity was measured with a conductivity meter (Eutech Instruments PC2700). The permeate flux ($J_{w}$) was then determined after the system reached a steady state within approximately 60 min. Subsequently, the DS was replaced with the second draw solution, 0.5 M NaCl aqueous solution, and the same parameters were obtained. The same procedure was repeated for the third and fourth concentrations of 0.75M and 1.0 M NaCl aqueous solutions, respectively. Using Equation (4), with the known $J_{w}$ and the difference in osmotic pressure (∆π) between FS and DS, A can be obtained, whereas B can be obtained through Equation (6) by determining the $J_{s}$ values beforehand via Equation (5), where VF0 is the initial volume of FS; Am is the membrane effective area; t is the length of time taken for the process; and CF and CF0 are the final and initial NaCl concentrations in the FS, respectively.
(4)$$J_{w}=A∆\pi$$
(5)$$J_{s}=\frac{C_{F}\left(V_{F0}\;{-\;J}_{W}A_{m}t\right)\;{-\;C}_{F0}V_{F0}}{A_{m}t}\;$$
(6)$$J_{w}=B∆C$$

Similar procedures were then repeated for the second stage with ALFS (active layer facing feed solution) orientation. The flux values and permeability values obtained from the previous stage were applied into Equations (7) and (8). In this equation, $\pi_{D,b}$ is the bulk osmotic pressure of the draw solution, $\pi_{F,b}$ is the feed osmotic pressure, D is the bulk diffusion coefficient of the DS, and k is the feed solute mass transfer coefficient. The determination of the S value was achieved using A, B and S as regression parameters by opting for the lowest global error as the reference, as the global error presents the minimum deviation between estimates and experimental data, as expressed in Equation (9). Figure 2 is provided to present a clear flow of the two-stage non-pressurized process. More details of the test protocol have been thoroughly described in [20,21,23,24].
(7)$$J_{w}=A\left\{\frac{\pi_{D,b}\exp\left(-\frac{J_{w}S}{D}\right){\;-\;\pi }_{F,b}\exp\left(\frac{J_{w}}{k}\right)}{1+\frac{B}{J_{w}}\left[\exp\left(\frac{J_{w}}{k}\right)\;-\;\exp\left(-\frac{J_{w}S}{D}\right)\right]}\right\}$$
(8)$$J_{s}=B\left\{\frac{C_{D,b}\exp\left(-\frac{J_{w}S}{D}\right)\;{-\;C}_{F,b}\exp\left(\frac{J_{w}}{k}\right)}{1+\frac{B}{J_{w}}\left[\exp\left(\frac{J_{w}}{k}\right)\;-\;\exp\left(-\frac{J_{w}S}{D}\right)\right]}\right\}\;$$
(9)$$E_{w\;}=\left(\frac{1}{n}\overset{n}{\underset{t=1}{\sum }}\sqrt{{\left(1\;-\;\frac{J_{w,i}^{C}}{J_{w,i}^{E}}\right)}^{2}}\right)100\%$$

## 3. Results and Discussion

### 3.1. Membrane Characteristics

Figure 3 shows the effect of the MAA UV grafted on the NF2 PES membrane surface water contact angle. Compared to the unmodified NF2 PES membrane having an average water contact angle of 63±3 °, all UV-grafted membranes were more hydrophilic, showing a water contact angle ranging from 38°–58°. For each MAA concentration, a longer irradiation time induces a lower water contact angle, rendering the membrane surface more hydrophilic [25,26,27,28]. This result disagrees with a previous study in which the authors concluded that it was not possible to confirm whether the irradiation time affects the hydrophilic character of the membrane, provided that only two samples out of five show improvement of the water contact angle [11]. Although it was found that the hydrophilic level increases with the increase in the MAA monomer concentration, the membranes 1.0MAA3 and 1.0MAA5 had higher water contact angle values than that of the membrane 0.5MAA10, which was prepared with a lower MAA concentration but a longer irradiation time. Similar behavior was detected for the membrane 0.5MAA3 compared to the membrane 0.1MAA10. This suggests that the increase in the MAA concentration promotes a thicker grafted layer, which results in pore plugging, making it difficult for water to be absorbed; hence, this leads to a higher water contact angle compared to applying a prolonged irradiation time, owing to chain scissions that encourage pore enlargement. The membrane 1.0MAA10, grafted with the highest MAA concentration and a long irradiation time, exhibited the highest hydrophilic character compared to all other membranes. It had a water contact angle 40% smaller than that of the NF2 PES membrane.

It is well known that the morphological roughness of the membrane surface also contributes to the measured contact angle (i.e., not only the surface chemistry but also the roughness, which is related to the porosity and pore size, affect the water contact angle measurements) [13,29,30]. This will be discussed later based on the AFM images’ analysis of the surface of the membranes.

Figure 4 shows the FTIR-ATR spectra of the unmodified NF2 PES and the UV-grafted membranes at different concentrations of monomers. The mechanism of photo-modification started with the absorption of UV light by the phenoxy phenyl sulfone chromophores in the backbone of the polymer, as illustrated in Figure 5. The absorption of high-energy radiation caused the hemolytic cleavage of a carbon-sulfur bond at the sulfone linkage, which generated two radical sites, aryl and sulfonyl. The produced radicals therefore induced the polymerization of methacrylic acid (MAA) at the reactive sites. In addition, the sulfonyl radical may also generate an additional aryl radical by losing sulfur dioxide, which may initiate polymerization with methacrylic acid. This characteristic band of the MAA monomer appears at 1714 cm^−1^ for the modified membranes, which is assigned to the carbonyl (C=O) groups [31]. The intensity of this peak increases considerably as the MAA concentration increases. Although a similar feature was spotted on 0.1 MAA3 regardless of its low intensity, a doublet at 1323 cm^−1^ and at 1294 cm^−1^ resulting from asymmetric O=S=O stretching of the sulfone group were still presence. In addition, strong aromatic C=C at 1487 cm^−1^ and 1504 cm^−1^ were also visible in the spectra, which suggests that the grafted monomer was not able to cover most of the membrane surface. However, these peaks underwent a reduction when the MAA concentration was 0.5 M and 1.0 M, implying that more monomers were successfully grafted on the membrane surface.

The FTIR spectra for the effect of the UV irradiation time are presented in Figure 6. A small transmittance of C=C at 1616 cm^−1^ for the NF2 PES membrane was observed to be gradually reduced for the modified membranes, implying that most vinyl bonds were crosslinked. The characteristic bands for COOH were spotted at 1714 cm^−1^ for all the modified membranes. However, it appeared that when the membrane was grafted for 10 min at a similar concentration, the intensity of the 1714 cm^−1^ bands was slightly reduced, indicating that less successful grafting had taken place than when the membrane was grafted for 5 min. A possible explanation for this might be that the prolonged irradiation time caused the grafted membrane to be de-grafted, and the trunk polymeric membrane deteriorated due to chain scission [12], hence reducing the intensity of the characteristic band. Full spectra and characteristic peaks for NF2 PES membrane and modified NF2 PES at different monomers concentrations and irradiation time are presented in Appendix A respectively.

### 3.2. Degree of Grafting (DG)

Figure 7 summarizes the results of the calculated degree of grafting (DG) of the modified membranes. It can be seen that the DG for the different sets of MAA concentrations resulted in different trends for the different irradiation times. For the lower MAA concentrations, 0.1 M and 0.5 M, the DG increased with the increase in irradiation time from 3 to 5 min, which aligned with the FTIR-ATR spectra, and it then decreased for a more prolonged irradiation time of up to 10 min. This suggests that the maximum UV irradiation time was reached, and no effective grafting took place; instead, more UV exposure started to break the monomers that had already been grafted on the NF2 PES membrane, resulting in a reduction in the DG [13]. This also explains why the membranes prepared with 1 M AA concentration experienced descending DG values when the UV−grafting time increased from 3 to 10 min. It must be emphasized that although the membranes 0.5MAA10 and 1.0MAA10 exhibited the lowest water contact angle values, their DG was not so high. Generally, the DG has a linear relationship with the water contact angle. This indicates the high roughness effect on the water contact angle measurement as a result of the overexposure of these two membranes to UV irradiation during grafting. Prolonged UV exposure may cause pore enlargement corresponding to chain scissions [32,33]. In addition, it is apparent that both the 1MAA3 and 1MAA5 membranes showed different water contact angles and DG values than the membrane 1MAA10, although the same MAA concentration was considered for UV grafting. For high monomer concentrations but limited exposure to UV irradiation, it is likely that the grafted MAA monomers partially block the pores and/or cause pore narrowing, which upsurges the reactions for homo-polymerization, resulting in an enhancement of the weight of the membrane and high DG values.

### 3.3. Surface Roughness

Atomic force microscopy (AFM) was used to analyze the surface morphology and roughness of the UV−grafted membranes and to compare the results with those of the unmodified NF2 PES membrane. The obtained roughness parameters (Ra, Rq, Rz) are summarized in Table 3. As can be seen, the NF2 PES membrane surface exhibits a ridge-and-valley surface morphology, typical of an interfacial polymerized PES membrane. The effect of the monomer concentration was presented through the roughness of 0.1 MAA3, 0.5 MAA3 and 1.0 MAA3.

It is apparent from Table 3 that the roughness increases initially with increasing monomer concentration from 0.1 M to 0.5 M and then decreases upon further increasing the concentration of MAA to 1.0 M at a constant UV exposure. Since the number of free radical sites on the membrane surface is the limiting factor for grafting, at a lower monomer concentration of 0.1 M, most of the monomers were effectively utilized by the available active sites on the membrane. Taking into consideration that the number of active sites is greater than the available monomers, increasing the monomer concentration would increase the deposition on the membrane surface; thus, a more even surface was obtained for 0.5 MAA3. On the other hand, 1.0 MAA3 has a higher surface roughness, most probably due to the higher concentration of monomers acting as barrier to the UV on the membrane surface, leading to less free radicals being generated from the UV irradiation, and so less grafting occurs. A positive correlation was found between the roughness parameters and the membrane surface, as presented in Figure 8. In contrast, the effect of UV irradiation times was different, as initially, after the irradiation took place for 3 min, 0.5 MAA3 showed a decrement in the average roughness. Based on Figure 9, when the irradiation of the UV was prolonged to 5 and 10 min, the mean surface roughness continued to increase, as did the Rq and Rz.

### 3.4. Membrane Structure

Figure 10 shows the FESEM images of the cross section for the membranes NF2 PES, 0.5 MAA3, 0.5 MAA5 and 0.5 MAA10. Apparently, a new layer was developed on the PES layer for all UV−grafted membranes (b,c,d). A slight increase in the selective layer of the UV−grafted membranes prepared with 0.5 M MAA concentration was detected with the increase in the UV irradiation time from 3 to 5 min (0.142 µm and 0.430 µm, respectively) as a result of the interaction between the monomers and the PES membrane, but this thickness of the selective layer became thin with a further increase in the UV irradiation time up to 10 min. This result agrees with the obtained DG values of these membranes. Similar behaviors were reported previously in other studies [34,35].

Figure 11 shows the effect of the UV irradiation time (3, 5 and 10 min) on the top layer of the modified membranes with 0.5 M and 1.0 M MAA concentrations. Differences can be detected from the top surface of both modified membranes with different MAA concentrations. The membrane surface became rougher when the UV irradiation time was increased, especially for 10 min (i.e., 0.5 MAA10 and 1.0 MAA10 membranes). This indicates a complete grafting and/or degradation caused by a prolonged UV irradiation time. The degradation through chain scission led to pore enlargement, therefore increasing the distance between the highest peak and the lowest valley on the membrane surface. These observations are also supported by the FESEM images of the membranes modified with 0.1–1.0 M MAA at 3 min UV irradiation time.

An increase in built-up monomer layers can be observed on the membrane surface based on Figure 12, in which the top images of the membranes are shown. Based on the top layer observations, a uniform grafted layer was discovered on the sample 0.5 MAA3 compared to 0.1 MAA3 and 1.0 MAA3. This result is quite unique, as previously, it has been reported that the surface roughness decreases with an increase in the monomer concentration [36,37]. The slight increase in texture on 1.0 MAA3 was likely due to pore enlargement that was caused by concentrated MAA monomers, which caused the monomer dissolution [17,38]. This assumption is rather controversial, as monomer is not the solvent for the PES membrane, yet it has been concluded by many researchers that this has led to membrane structure deterioration [13,28]. In contrast, according to Seman et al. [18], less grafting on the membrane surface when a high concentration of monomers was considered was due to the repulsive force between carboxylic acid and the sulfonyl group of the PES material. In addition, it was claimed that even with low grafting, chain scission still governs the membrane pore enlargement, which is contradictory to the obtained results in the present study.

### 3.5. Evaluation of FO Membrane Performance

The obtained FO water permeate flux (J_w_) and the reverse salt flux (J_s_) are plotted in Figure 13 for the unmodified NF2 PES membrane and all UV−grafted membranes. As can be seen, all modified membranes exhibit higher J_w_ and J_s_ values than those of the NF2 PES membrane. For each MAA concentration, J_w_ increases with the increase in the UV irradiation time. A nearly similar behavior is exhibited by J_s_. In contrast, no clear behavior can be detected for the effect of the MAA concentration on J_w_ and J_s_ values when maintaining the same UV irradiation time. These results may be attributed partly to the membrane thickness, pore size and porosity, and in general to the structural parameter S. This will be explained later on.

A higher MAA concentration provides more progressive monomer depositions on the membrane surface to form the active layer, but it causes pore plugging, therefore reducing J_w_ values, as shown for the membranes 1.0 MAA3 and 1.0 MAA5 compared to the membranes modified with 0.5 M. It is known that the polymerization rate increases for high concentrations of monomers, resulting in an enhanced viscosity that would obstruct the diffusion of monomers in the membrane. In addition, a high viscosity also hinders the UV from reaching the membrane surface [39], inducing less free radicals to be generated for the grafting process, and consequently resulting in a low degree of grafting and higher J_w_ values, as occurred for the membrane 1.0 MAA10. On the other hand, for the case of prolonged grafting time, the chain scission mechanism is the most dominant cause of the enhancement in J_w_ values due to pore enlargement during UV irradiation time, as occurred for the membrane 1.0 MAA10. Another piece of evidence for this claim is associated with all samples modified for 10 min, as the corresponding membranes showed a significant enhancement of both J_w_ and J_s_. It is important to highlight that pore plugging did not occur at low MAA concentrations and for long irradiation times. Therefore, at a lower concentration of monomers but a longer irradiation time, higher solute and water fluxes are achieved due to membrane degradation caused by a prolonged exposure to UV light [40]. Typically, the two mechanisms of the crosslinking of MAA and the chain scission of the PES backbone compete together during UV grafting. Basically, increasing the time of UV exposure from 3 to 5 min accelerates the deposition of monomers on the membrane surface, specifically for 0.1 M and 0.5 M MAA concentrations. However, when the irradiation time increases to 10 min, chain scission starts to be the dominating mechanism. This phenomenon is possible, considering how sensitive the polyethersulfone polymer is towards the UV irradiation.

According to the previous literature [28], water flux is reduced when the MAA concentration is increased up to the optimum level. It is assumed that the number of monomers deposited on the membrane surface increases with the increase in MAA concentration, resulting in more coverage for grafting on the membrane surface, leading to a permeate flux decline. In the present work, for a higher MAA monomer concentration, the applied UV energy is unlikely to reach the membrane surface easily, promoting less MAA monomer crosslinking with the PES, and it is almost impossible for chain scission to occur. The observed enhanced J_w_ and J_s_ with the increase in the MAA concentration may be due partly to deterioration of the membrane structure. This deterioration theory has been widely investigated previously in pressurized membrane processes [12,38,41].

### 3.6. Intrinsic Test

As indicated earlier, the water permeability (A), solute permeability (B) and structural parameter (S) were determined by a two-stage method. The first stage was conducted using the ALDS orientation under four different NaCl concentrations of the draw solution (0.5 M, 0.75 M, 1.0 M and 1.25 M) in order to obtain the A and B values. The second stage was conducted using the ALFS orientation at similar NaCl concentrations to determine the S value using A, B and S as regression parameters with the objective function of minimizing the error. The highest E value recorded was 8.58%, which is still considered reasonable, only because the S value for NF2 PES cannot be determined via this method due to the inadequate amount of the obtained J_w_ during the first stage of the experiment [20]. As it was designed for the NF separation process, the NF2 PES membrane was not practically suitable for the FO process due to its relatively low hydrophilic character and the necessary application of a transmembrane hydrostatic pressure to operate, particularly in the ALDS orientation. The obtained values of A, B and S for both the unmodified and the UV−grafted membranes are summarized in Table 4, not considering the S value for the pristine membrane NF 2 PES.

Another parameter that was successfully enhanced after membrane modification is the water permeability (A). It is obvious that the A values for all modified membranes were larger than that of the NF2 PES membrane, proving that UV grafting can improve the permeability of the membrane for the FO process. As can be seen in Figure 14, generally, the A and B values increased, whereas the S value was reduced with the enhancement of the grafting time. This trend was observed for all concentrations of MAA.

In general, a larger S value results in a higher internal concentration polarization (ICP) [42]. Therefore, it is advantageous to have a lower S parameter in order to reduce the ICP factor, increasing the A value and decreasing the B value. It is worth noting that for an appropriate FO separation process, the membrane should exhibit a highly selective active layer (i.e., high A and low B values) with a lower structural parameter of the membrane support layer [43]. For instance, the membrane 0.5 MAA5 shows the lowest S value (637.69 µm) and B/A ratio (10.9). A smaller B/A ratio indicates less severe reverse solute diffusion (RSD) from the draw solution (DS; NaCl) side into the feed solution (FS; DI water) side during the FO process [44,45].

## 4. Conclusions

The commercial NF2 PES membrane was modified by UV grafting using different MAA concentrations (0.1 M, 0.5 M and 1.0 M) for three different exposure times (3, 5 and 10 min). The FO performance of the UV−grafted membranes was investigated together with their intrinsic properties. This study reveals evidence of the improvement of the hydrophilic character of the top surface of the MAA-grafted membranes. The NF2 PES membrane had an average contact angle of 63° ± 3.5, which was more hydrophobic than the modified membranes. After UV grafting, the water contact angle was reduced from 63° ± 3.5 for the NF2 PES membrane to 32° ± 1.6 − 58° ± 2.9 for the UV-grafted membranes. For low MMA concentrations, the DG increased with the increase in the irradiation time from 3 to 5 min, which aligned with the FTIR-ATR spectra, and it then decreased for a more prolonged irradiation time up to 10 min, suggesting that the maximum UV irradiation time was reached at 5 min. The AFM results show that with lower MAA concentrations and a shorter irradiation time, a rougher membrane surface is formed due to incomplete surface grafting, considering that the active radicals are the limiting factor. The modified membranes appeared to have larger water flux (J_w_) and solute flux (J_s_) than the NF2 PES membrane. This suggests that the UV−grafting modification approach has a comparable effect in the FO separation process and in the pressurized filtration process. The observed fluxes show that the longer irradiation time allowed for chain scission to occur more than the crosslinking that eventually degraded the membrane material, and higher concentrations promote pore plugging. The structural parameter S decreased with the increase in the grafted membrane’s permeability (A, B parameters). All grafted membranes exhibited a lower S value than that of the commercial NF2 PES membrane. The modified membrane 0.5 MAA5 showed the lowest S value (637.69 µm) and B/A ratio (10.9).

## Figures and Tables

**Figure 1 membranes-13-00232-f001:**
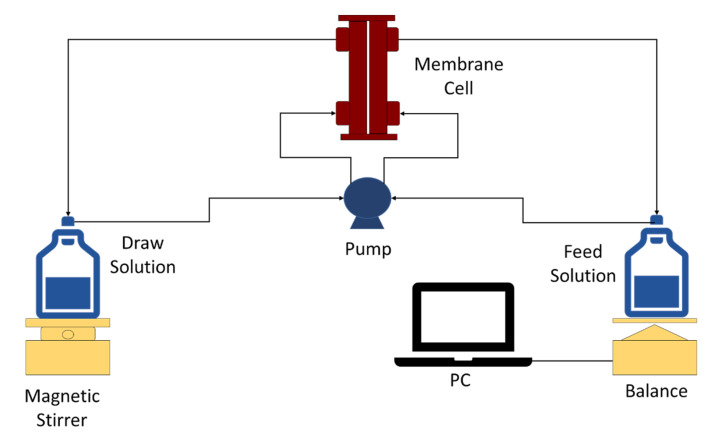
Schematic diagram of FO system.

**Figure 2 membranes-13-00232-f002:**
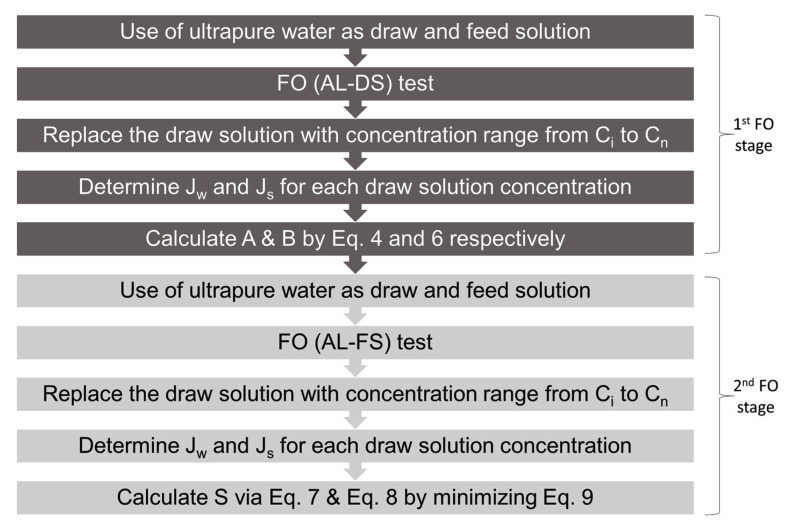
Flow chart of the two-stage non-pressurized method.

**Figure 3 membranes-13-00232-f003:**
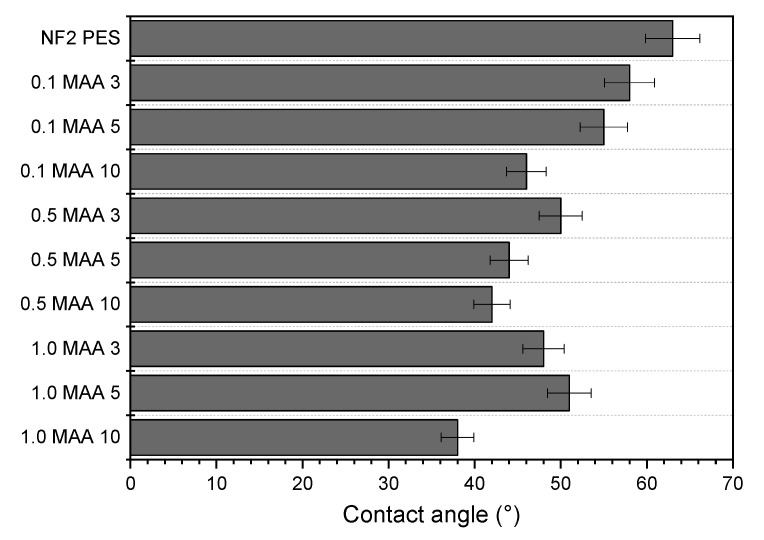
Water contact angle of the unmodified and UV-grafted membranes.

**Figure 4 membranes-13-00232-f004:**
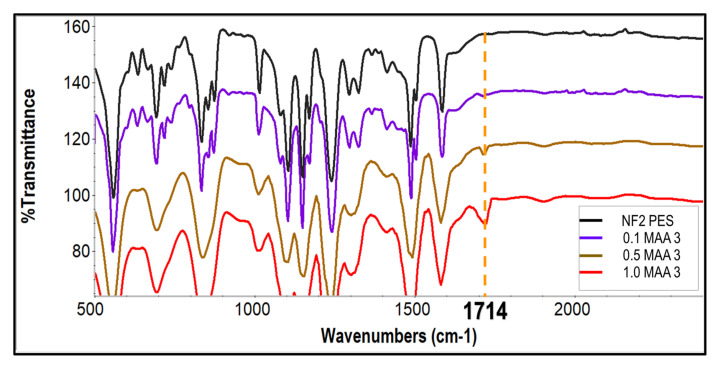
FTIR-ATR spectra of the unmodified NF2 PES and UV-grafted membranes 0.1 MAA3, 0.5 MAA3 and 1.0 MAA3.

**Figure 5 membranes-13-00232-f005:**
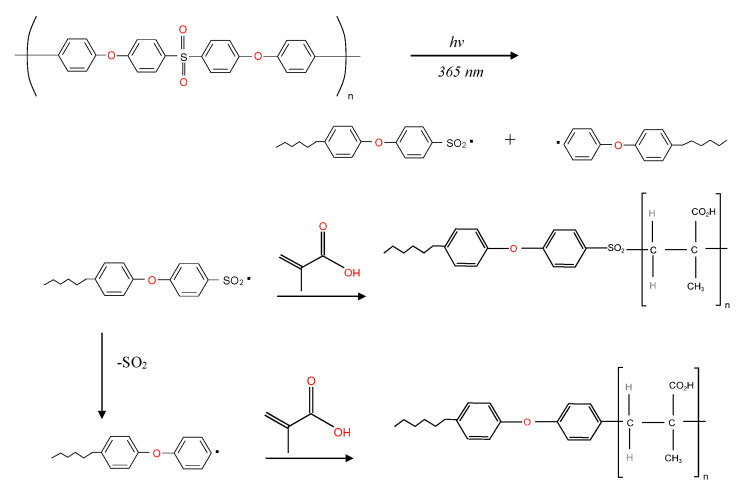
UV−photochemical grafting modification of polyether(sulfone) with methacrylic acid.

**Figure 6 membranes-13-00232-f006:**
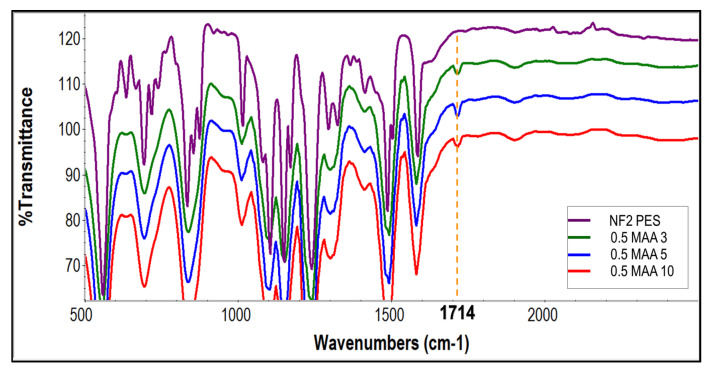
FTIR-ATR spectra of the unmodified NF2 PES and UV−grafted membranes 0.5 MAA3, 0.5 MAA5 and 0.5 MAA10.

**Figure 7 membranes-13-00232-f007:**
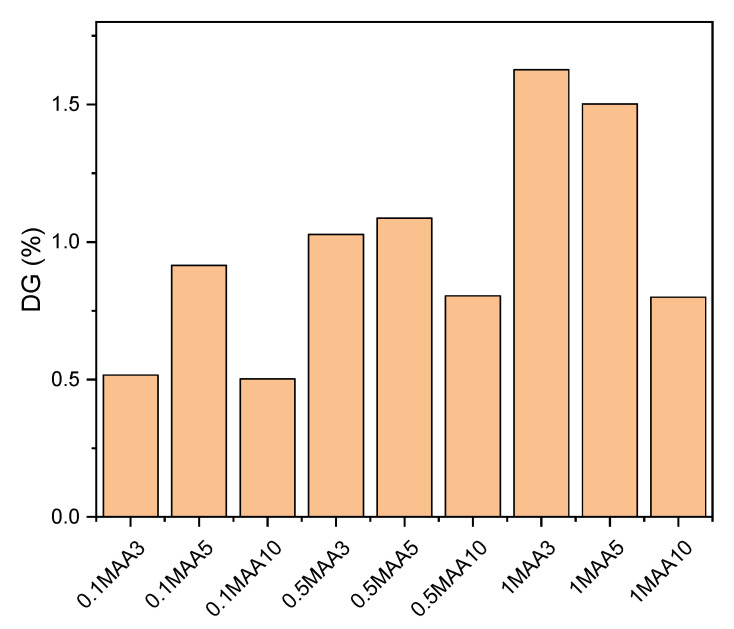
Degree of grafting of the modified membranes.

**Figure 8 membranes-13-00232-f008:**
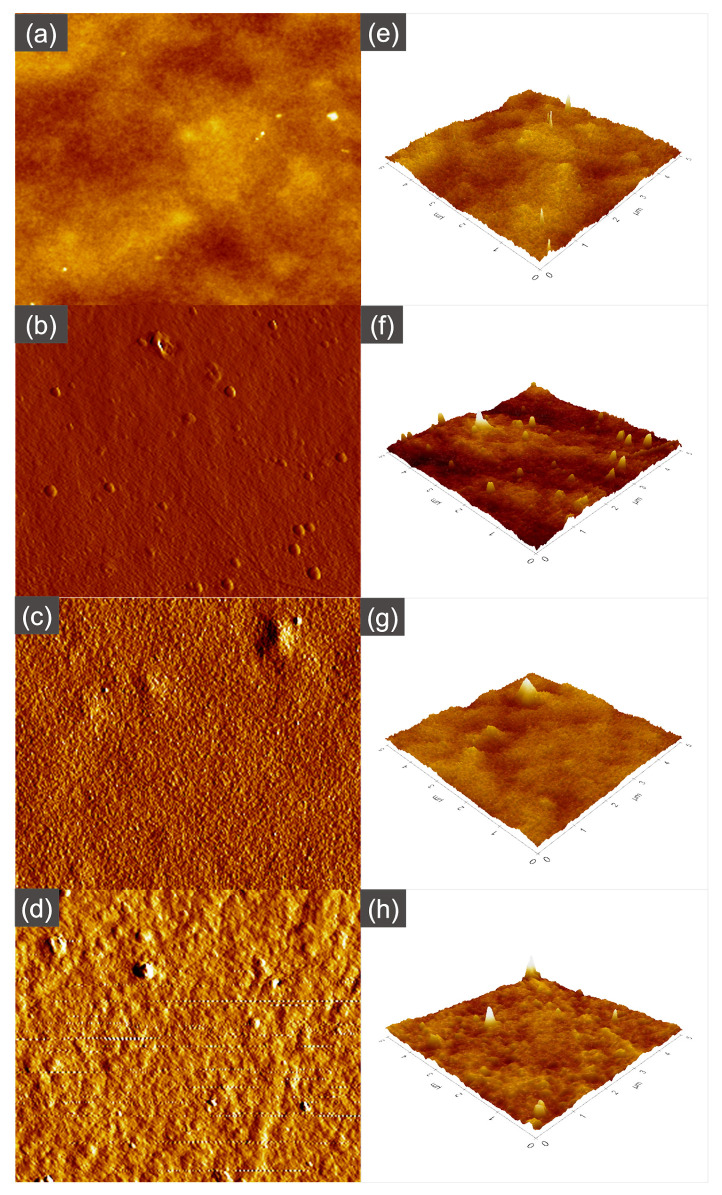
AFM images (5 µm × 5 µm) of the top surface of the NF2 PES membrane (**a**,**e**) and UV−grafted membranes 0.1 MAA3 (**b**,**f**), 0.5 MAA3 (**c**,**g**) and 1.0 MAA3 (**d**,**h**).

**Figure 9 membranes-13-00232-f009:**
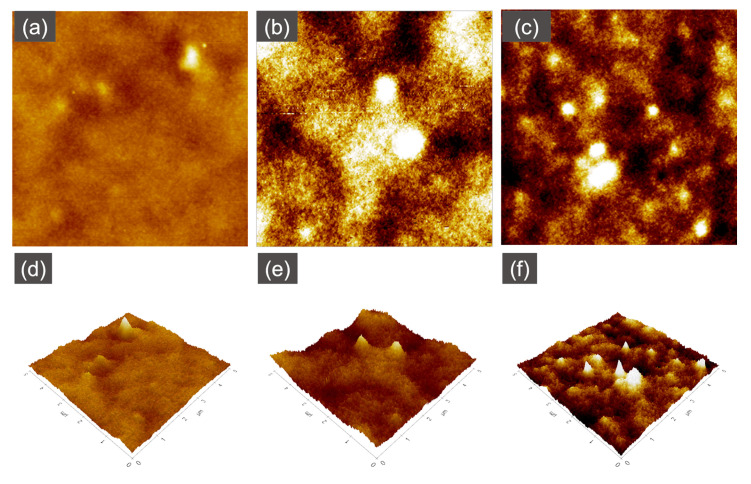
AFM images (5 µm × 5 µm) of the top surface of the modified membranes 0.5 MAA3 (**a**,**d**), 0.5 MAA5 (**b**,**e**) and 0.5 MAA10 (**c**,**f**).

**Figure 10 membranes-13-00232-f010:**
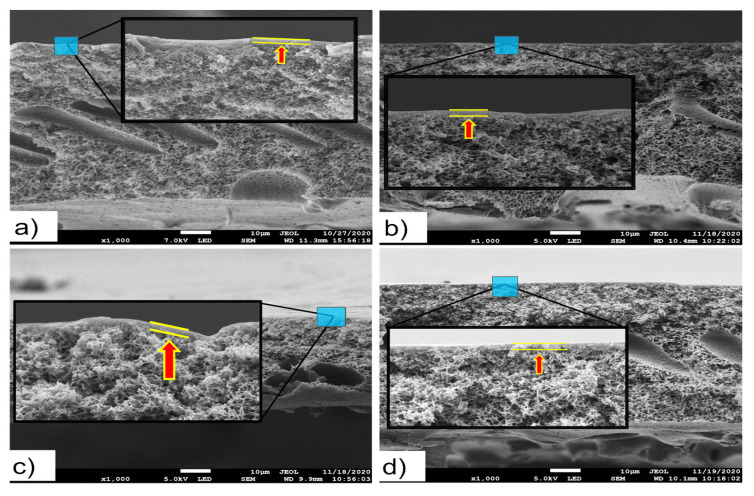
FESEM images (1000× magnification) of the membranes: (**a**) cross section of NF2 PES, (**b**) cross section of 0.5 MAA3, (**c**) cross section of 0.5 MAA5, (**d**) cross section of 0.5 MAA10.

**Figure 11 membranes-13-00232-f011:**
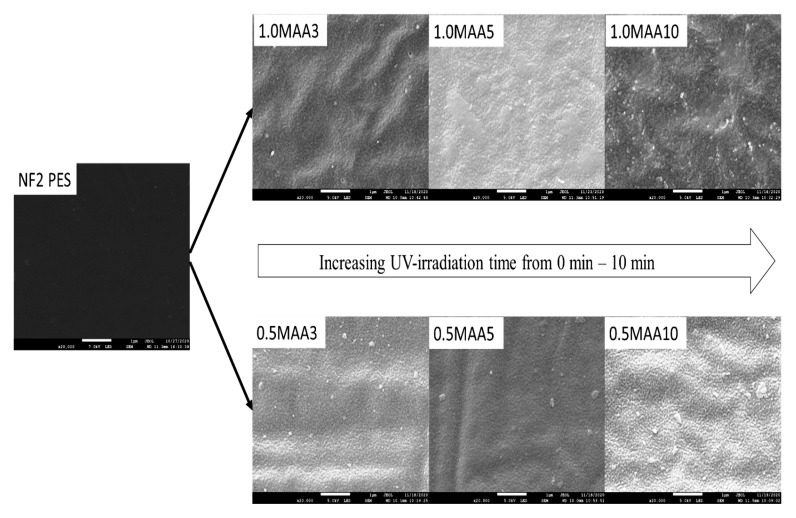
FESEM images (20,000× magnification) of the top membrane surface showing the effect of UV irradiation time.

**Figure 12 membranes-13-00232-f012:**
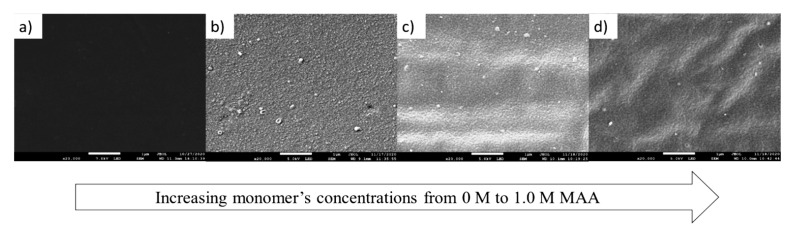
FESEM images (20,000× magnification) of the top surface of the membranes: (**a**) NF2PES, (**b**) 0.1 MAA3, (**c**) 0.5 MAA3 and (**d**) 1.0 MAA3.

**Figure 13 membranes-13-00232-f013:**
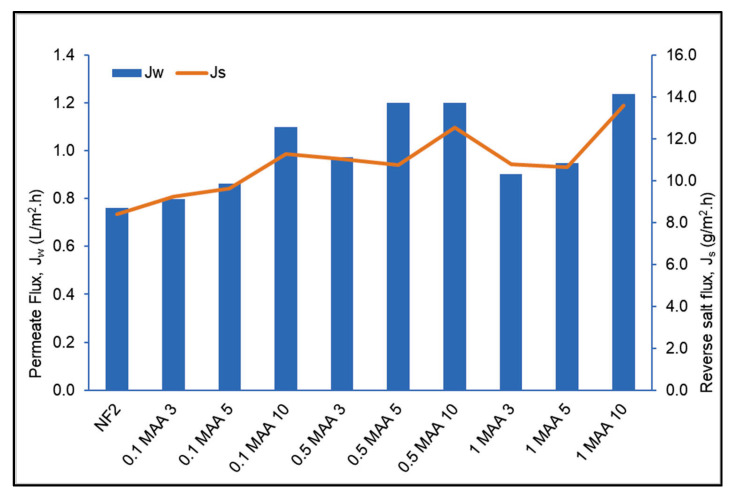
Water permeate flux (J_w_) and reverse salt flux (J_s_) of the unmodified NF2 PES membrane and the UV−grafted membranes under different modification conditions, with 1 M NaCl draw solution and DI water as feed.

**Figure 14 membranes-13-00232-f014:**
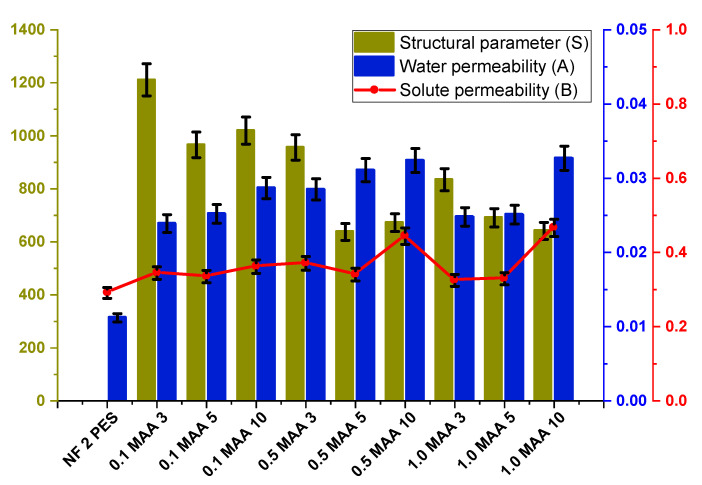
Structural parameter (S), water permeability (A) and salt permeability (B) of the UV−grafted membranes.

**Table 1 membranes-13-00232-t001:** Summary of research studies on the UV-grafted polyether sulfone (PES) membrane for FO, nanofiltration (NF) and ultrafiltration (UF) applications.

Membrane	Monomers	Remarks	Ref.
NF PES membrane purchased from Amfor Inc. (China)	AA	For a high monomer concentration, the thickness of the grafted layer increases, improving the transport resistance.	[15]
UF PES membrane purchased from Amfor Inc. (China)	AA	The effect of degradation of PES due to irradiation is unavoidable, because prolonging grafting time leads to an increase in the pore size due to polymeric chain scission.	[11]
PES UF membrane manufactured by DSS	NVP, 2-acrylamido glycolic acid monohydrate (AAG) and 2-acrylamido-2-methyl-1-propane sulfonic acid (AAP)	The membrane grafted with AAG shows a higher grafting level compared to the rest of the membranes.	[16]
PES membranes with different MWCO obtained from Pall Filtron Corp. (East Hills, NY)	NVP, 2-hydroxyethyl methacrylate (HEMA), 2-acrylamido-2-methyl-1-propane sulfonic acid (AMPS), 3-sulfopropyl methacrylate potassium salt (SPMA), 2-acrylamido glycolic acid (AAG) and AA	The selection of the monomer is dictated by the applications in which the modified membrane would be used.	[17]
NFPES10, supplied by Hoechst Company	AA	The AA UV-grafted membrane exhibits higher rejection factors and low fouling tendency versus the unmodified membrane for similar humic acid (HA) pH values.	[18]
PES flat sheet membranes (self-made via phase inversion)	2-hydroxyethyl methacrylate (HEMA), AA, 1,3-phenylenediamine (mPDA) and ethylenediamine (EDA)	All grafted PES membranes are more hydrophilic than unmodified ones, and HEMA-grafted membrane exhibits the best hydrophilic surface.	[19]
NFPES10, supplied by Hoechst Company	NVP and AA	Higher degree of grafting (DG) is observed for longer irradiation until a certain level at which it starts to decrease, probably due to overexposure to UV light.	[14]

**Table 2 membranes-13-00232-t002:** Prepared membranes and modification conditions.

Membrane	Monomer Concentration (M)	Irradiation Time (min)
NF2 PES	-	-
0.1MAA3	0.1	3
0.1MAA5	0.1	5
0.1MAA10	0.1	10
0.5MAA3	0.5	3
0.5MAA5	0.5	5
0.5MAA10	0.5	10
1.0MAA3	1.0	3
1.0MAA5	1.0	5
1.0MAA10	1.0	10

**Table 3 membranes-13-00232-t003:** Roughness parameters (Ra, Rq, Rz) of the surface of the unmodified and the UV−grafted membranes.

Membrane	Ra (nm)	Rq (nm)	Rz (nm)
NF2 PES	2.173	2.769	24.429
0.1 MAA 3	3.276	4.304	36.371
0.5 MAA 3	1.502	1.827	10.915
1.0 MAA 3	2.304	3.200	51.720
0.5 MAA 5	1.629	2.073	18.826
0.5 MAA 10	2.237	3.244	35.234

**Table 4 membranes-13-00232-t004:** Intrinsic characteristics (A, B, S) of the unmodified and UV−grafted membranes.

Sample	A (Lm^−2^h^−1^bar^−1^)	B (Lm^−2^h^−1^)	S (µm)	B/A	E (%)
NF 2 PES	0.0112	0.2912	-	26.0	8.58
0.1 MAA 3	0.0239	0.3445	1211.23	14.4	1.41
0.1 MAA 5	0.0252	0.3352	965.85	13.3	1.21
0.1 MAA 10	0.0287	0.3620	1019.96	12.6	1.09
0.5 MAA 3	0.0285	0.3707	956.12	13.0	0.90
0.5 MAA 5	0.0311	0.3404	637.69	10.9	1.11
0.5 MAA 10	0.0324	0.4441	672.45	13.7	1.44
1.0 MAA 3	0.0248	0.3248	834.80	13.1	1.52
1.0 MAA 5	0.0251	0.3295	690.90	13.1	1.11
1.0 MAA 10	0.0327	0.4664	641.62	14.3	1.52

## Data Availability

The data presented in this study are available on request from the corresponding author.

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
