# Peer review of "Effect of Methacrylic Acid Monomer on UV-Grafted Polyethersulfone Forward Osmosis Membrane"

_membranes, 2023, doi:10.3390/membranes13020232_

Round 1

Reviewer 1 Report

The present study is about the investigation of the FO performance of the UV-grafted NF2 PES membrane using the monomer MAA. The effects of the MAA concentration and irradiation time on the characteristics and FO performance of the modified membranes have been investigated. Comparisons have been carried out between the modified and unmodified NF2 PES membranes. The manuscript is well written and can be accept for publication after addressing the following comments:

Language:

The manuscript should be checked for grammatical revision.

Tables and figures:

Table 01 is numbered two times and because of this all tables numbering are wrong. This must be corrected and in the text it must be correctly cited.

On line 284 page 11, the table number in the text is missing.

The authors should thoroughly check such mistakes in the remaining manuscript.

Results and discussion:

Line 177 to 179 must be deleted.

In the results and discussion section line 187, the authors disagree with the previous study, about the effect of irradiation time on the hydrophilic character of the membrane. The authors should explain that why that was not possible to compare the irradiation time on the hydrophilic character of the membrane in the previous study. The authors may explain it here or in the introduction/literature review section.

Conclusions:

Lines 460 to 464: the sentence,  

“For low MMA concentrations, the DG increases with the increase of the irradiation time from 3 to 5 min, which is aligned with the FTIR-ATR spectra, and then decreases for more pro- longed irradiation time up to 10 min suggesting that the maximum UV-irradiation time was reached.”

Should be revised as

“For low MMA concentrations, the DG increases with the increase of the irradiation time from 3 to 5 min, which is aligned with the FTIR-ATR spectra, and then decreases for more pro- longed irradiation time up to 10 min, suggesting that the maximum UV-irradiation time was reached at 05 min.”

Reviewer 2 Report

This paper reported UV graft polymerization of MAA on the PES membranes toward a high performance FO system. Some issues should be addressed.

1.     The reaction mechanism of MAA on the PES membrane should be presented, such as the reaction process.

2.     It is not suitable for calculating the degree of grafting via just the weight change. In addition, the contact angel is not agreed with the DG, why?

3.     The scale bar of AFM image should be presented.

4.     The SEM images are not clear, especially for the scale bar.

5.     The UV grafting of MAA increase the JW and JS, where the increase of JS leads to the negative effect on FO system.

6.   More relevant works should be cited in the introduction section, such as other sustainable materials for FO membrane. (International Journal of Biological Macromolecules, https://doi.org/10.1016/j.ijbiomac.2022.12.052; Carbohydrate Polymers, https://doi.org/10.1016/j.carbpol.2022.119601)

Reviewer 3 Report

 Reviewer’s comments:

This study discusses effect of methacrylic acid monomer on the UV-grafted poly-ethersulfone forward osmosis membrane. There are some questions.

Details are as shown below.

1. Are there any corresponding references for Line 47~53? 

2. Some references are poorly correlated, for example, the first reference in Line 66. Please check again for similar problems.

3. The infrared spectra in Figure 4 and Figure 5 are incomplete. The characteristic peaks are not clear and hard to quantify the grafting reaction. In order to further characterize the grafting reaction, please supplement the x-ray photoelectron spectroscopy (XPS) measurements.

4. Line 286~289, how to characterize the number of free radical sites on the membrane surface?

5. It’ reported that in graft polymerization technique, there are competitions between effective grafting and chain scission whereby only one mechanism will dominate at a time. Please analysis the mechanism of methacrylic acid monomer on the UV-grafted poly-ethersulfone forward osmosis membrane in detail under different reaction conditions.

6. Line 343~346, “ The increase of build-up monomer layers can clearly be spotted on the membrane surface based on Figure 11 where top images of the membranes were shown. The increase of build-up monomer layers is not obvious.

7. What are the stability and anti-fouling properties of UV-grafted poly-ethersulfone forward osmosis membrane?

Round 2

Reviewer 2 Report

Authors have addressed the issues.

Reviewer 3 Report

 There are still some questions about the manuscript. Comments are shown as below.

1. Please supplement the complete IR spectra and the characteristic peaks should be marked in detail. 

2. Line 233-235, it is difficult to keep the thickness and transparency consistent in infrared test, therefore, the quantitative analysis in the manuscript should be more accurate. Can internal standard method be used for the quantitative analysis?
